# Smartphone-Based Visual Inspection with Acetic Acid: An Innovative Tool to Improve Cervical Cancer Screening in Low-Resource Setting

**DOI:** 10.3390/healthcare10020391

**Published:** 2022-02-18

**Authors:** Jana Sami, Sophie Lemoupa Makajio, Emilien Jeannot, Bruno Kenfack, Roser Viñals, Pierre Vassilakos, Patrick Petignat

**Affiliations:** 1Department of Pediatrics, Gynecology and Obstetrics, Geneva University Hospitals, 1205 Geneva, Switzerland; lemoupasophie@yahoo.fr (S.L.M.); pierrevassilakos@bluewin.ch (P.V.); patrick.petignat@hcuge.ch (P.P.); 2Bafoussam Regional Hospital, Bafoussam FCQ5+X4, Cameroon; 3Institute of Global Health, Faculty of Medicine, 1211 Geneva, Switzerland; emilien.jeannot@unige.ch; 4Addiction Medicine Service, Lausanne University Hospital (CHUV), 1011 Lausanne, Switzerland; 5Department of Gynecology-Obstetrics, Faculty of Medicine and Pharmaceutical Sciences, University of Dschang, Dschang C2WW+4M, Cameroon; brunokenfack@gmail.com; 6Signal Processing Laboratory (LTS5), École Polytechnique Fédérale de Lausanne (EPFL), 1015 Lausanne, Switzerland; roser.vinalsterres@epfl.ch; 7Geneva Foundation for Medical Education and Research, 1202 Geneva, Switzerland

**Keywords:** cervical cancer screening, low and middle-income countries, training, VIA/VILI, smartphone-based, digital colposcopy, artificial intelligence

## Abstract

Visual inspection with acetic acid (VIA) is recommended by the World Health Organization for primary cervical cancer screening or triage of human papillomavirus-positive women living in low-resource settings. Nonetheless, traditional VIA with the naked-eye is associated with large variabilities in the detection of pre-cancer and with a lack of quality control. Digital-VIA (D-VIA), using high definition cameras, allows magnification and zooming on transformation zones and suspicious cervical regions, as well as simultaneously compare native and post-VIA images in real-time. We searched MEDLINE and LILACS between January 2015 and November 2021 for relevant studies conducted in low-resource settings using a smartphone device for D-VIA. The aim of this review was to provide an evaluation on available data for smartphone use in low-resource settings in the context of D-VIA-based cervical cancer screenings. The available results to date show that the quality of D-VIA images is satisfactory and enables CIN1/CIN2+ diagnosis, and that a smartphone is a promising tool for cervical cancer screening monitoring and for on- and off-site supervision, and training. The use of artificial intelligence algorithms could soon allow automated and accurate cervical lesion detection.

## 1. Introduction

Cervical cancer (CC) is one of the most common cancers in women and one of the leading causes of cancer death in women in developing countries, although it is almost a totally preventable cancer. In 2020, more than 600,000 new cases of cervical cancer were reported worldwide [1]. To reduce the burden of disease, the World Health Organization (WHO) has launched a global program to eliminate cervical cancer, with the following targets: (i) all countries have to achieve 90% vaccination coverage, (ii) 70% of screening coverage, and (iii) 90% access to treatment for cervical pre-cancer or cancer [2].

To reach the second and third WHO targets, towards cervical cancer elimination, women should be screened using high-performance tests between the ages of 5 to 45 years old, coupled with treatment, if required [3]. Visual assessment of the cervix with acetic acid (VIA) has been adopted by the WHO in primary screenings or in the triage of HPV-positive women as an affordable and adapted method in low-resource settings [3]. However, “naked-eye” VIA assessment remains a highly subjective test with low performance and limited quality control [4]. Limitations were first inherent to the technique itself, but variations of training, mentorship, and supervision may also account for the performance difference observed across studies.

To provide easier recognition of cervical intraepithelial neoplasia (CIN) by screeners, digital cervical pictures taken during VIA (D-VIA) with a camera have progressively developed and are being adopted by many with encouraging results [5,6]. However, picture acquisition has not been so easy to perform; in particular, good image quality, managing difficulties (e.g., focusing), and adequate light sources have been challenging.

To effectively evaluate the cervix before and after acetic acid application, light and magnification are needed, which can be easily and adequately obtained with a smartphone device. Smartphones can assist screeners in determining morphology, localization, and the type of transitional zone, and distribution of aceto-whitening features. Implementing smartphone D-VIA has opened up a new dimension to VIA and may be a major improvement in cervical cancer prevention. Furthermore, the increasing prevalence of smartphone-use in low-resource settings makes it an ideal low cost device. Research teams and engineers have developed programs to increase accuracy of D-VIA diagnosis from smartphone pictures. Our aim was to provide an overview of the data available on smartphone-use for cervical cancer screening in low-resource settings and describe its promises and challenges.

## 2. Methodology

A literature search was conducted on MEDLINE and LILACS, selecting English- and French-written studies published between January 2015 and November 2021. Terms used for research were: (cervical cancer screening) and ((smartphone) or (digital colposcopy)). Articles were excluded from the title, abstract, and/or methodology: (i) if they had not been conducted in a cervical cancer screening context using D-VIA, (ii) if they did not include the use of smartphones for D-VIA image acquisition, (iii) if the study was not conducted in a low-resource setting or low and middle income countries (LMICs), and (iv) if the study data were not published in full articles. Specific MeSH terms (VIA, D-VIA, LMICs) were voluntarily not used because it limited greatly the number of results. Research period was chosen arbitrarily to overview most recent data. We also summarized the articles published on artificial intelligence (AI) according to our research for cervical pre-cancer identification. These studies were not necessarily conducted in low-resource settings considering that AI could be used in the near future in LMICs (Figure 1). Reference lists of identified papers were also reviewed to ensure that all studies meeting our inclusion criteria were considered. Two reviewers independently assessed each article for eligibility.

## 3. Visual Inspection with Acetic Acid (VIA): Strengths and Limitations

VIA is a procedure in which a healthcare provider applies a solution of 3 to 5% acetic acid with the aim of highlighting and identifying CIN. The procedure requires an experienced screener able to conduct a naked-eye examination and interpret the cervical change before and after application of acetic acid. VIA interpretation is crucial for determining if the screen is positive or negative and to decide the strategy of treatment, if positive. The decision to treat relies on the provider’s evaluation and experience; false positive cases may lead to overtreatment while false negatives will lead to misdiagnosis. If the screen is positive and if treatment is required, the provider must decide if the condition is eligible for ablative treatment (thermal ablation or cryotherapy) or excisional treatment (i.e., large loop excision of the transformation zone (LLETZ)) or referral to multimodal therapy in case of suspicion of invasive cancer. The main strength of the VIA approach is that it is affordable and offers the possibility of immediate results and treatment in a single visit. 

Since the end of the 1990s, some countries have endorsed VIA instead of cytology as a primary screening in their national cervical cancer control programs, linking in the same visit screenings and treatments [7]. However, the technique is considered by some as a low standard of care, with critical weaknesses, such as its subjectivity, which leads to high variability in the provider’s performance as well as a lack of validated quality assurance [8]. 

Mustafa et al. conducted a systematic review to compare the accuracy of an HPV test, cytology (cervical smear), unaided VIA, and a colposcopy for cervical cancer screening in high-income countries. Results showed a pooled sensitivity of 69% (CI 95% 54–81) for VIA compared to 95% (CI 95% 84–98) for HPV testing; and a specificity of 87% (CI 95% 79–92) for VIA compared to 84% (CI 95% 72–91) for HPV. When compared to cervical smear accuracy, VIA caused a significant increase in overtreatment with 58 more false positives for 1000 women [9]. These weaknesses might explain the absence of implementation of VIA-based screening programs more than 20 years after publication of the WHO guidelines [10]. Despite the existence of recommendations and important investments made by private and public organizations in the field, difficulties have occurred with implementation of the Sub-Saharan cervical screening program [11]. This is of great concern, as the likelihood of reducing the incidence of cervical cancer relies on an effective and inexpensive screening method and a well-organized program.

## 4. VIA-Enhanced with Digital Imaging

The advent of digital cervical photography after acetic application (D-VIA), taken by on-site healthcare providers with cameras in order to assist the CIN identification was an important step in cervical cancer screenings [12]. A study evaluating cervical digital photography and colposcopy by different observers reported an agreement in 89.9% of the cases (kappa (k) = 0.588), a higher sensitivity (52.5%), and positive predictive value (PPV) (60%) as compared to colposcopy (35% and 48.28%, respectively). Specificity (91.8% vs. 91.2%), negative predictive value (NPV) (89.3% vs. 85.8%), and diagnostic accuracy (84.4% vs. 80.7%) were quite similar. This study supports that cervical digital photography may be an alternative to colposcopy for CIN diagnosis [13]. In Zambia and Kenya, cervix digital images were taken by a camera and it was reported difficult to capture images and retain details without distortion (fluctuation in color, not enough light intensity, loss of resolution) [14,15]. Ensuring image quality (color accuracy, focus) with a camera is a challenging issue.

Alternatively, smartphones, which are often combined with auxiliary lenses (i.e., MobileODT), allowing the acquisition of high-resolution cervical images, enable visualization of morphological features, which may be difficult to see with naked-eye alone [16]. Advantages of a smartphone over a traditional camera is its ease of use, it does not need an external light source, and it allows easy zooming and comparisons of different pictures taken during an exam (native, VIA). D-VIA seems to have a higher discriminative power when compared to a naked-eye examination in detecting precancerous lesions; thus, making it an additive value to traditional VIAs to improve the diagnosis of cervical precancerous lesions.

## 5. Performance for CIN2+ Diagnosis

Image quality for cervical intraepithelial neoplasia (CIN) detection relies on the quality of the digital technology used, as well as on image classification and registration. Currently, there is no standard specific recommendation similar to what exists in other medical specialties (i.e., digital imaging and communication in medicine (DICOM)).

In Madagascar, Gallay et al. evaluated the quality of smartphone images to assess feasibility and usability of a mobile application in low-resource settings. Women aged 30–65 years old were recruited in a cervical cancer screening campaign and HPV-positive ones underwent VIA assessment [17]. Pictures were taken using a Samsung Galaxy S5 and a phone application called “Exam” was used to classify images. A total of 208 consecutive pictures were assessed by observers and quality was judged as adequate for diagnoses in 93.3% of cases.

Tran et al. reported a sensitivity of 71.3% (95% CI 67–75.7) and a specificity of 62.4% (95% CI 57.5–67.4) for CIN2+ detection from D-VIA images taken by smartphones—a Samsung Galaxy S4 and S5—in Madagascar and evaluated by off-site gynecologists [18].

Studies that evaluated these issues, with or without image management applications, support that the quality of the image was, most of the time, considered sufficient for diagnosis and the decision of treatment. However, current evidence regarding the use and benefits regarding implementation of digital-VIA for CIN2+ diagnosis is still weak, as there are no randomized controlled trials (VIA versus D-VIA) or large prospective studies that have evaluated this issue. Results show overall good specificity for D-VIA; sensitivity values are however heterogeneous (Table 1).

## 6. On-Site Training and Supervision

Current understanding of a VIA-based approach supports that the method needs to be conducted with adequate training, supervision, and quality control to optimize the technique [8]. In a “screen and treat” strategy, frontline screeners play key roles and supervision is important, but it may not be available in healthcare centers located in remote areas [24]. Production of digital images allow to have records of the appearance of the cervix before and after acetic acid application, which permits screeners and supervisors to review the selected cases for quality control.

Asgary et al. explored the acceptability and feasibility of smartphone-based training of Ghanaian healthcare professionals using VIA and D-VIA. Community health nurses completed a two-week on-site introductory training, followed by ongoing, three-month text messaging, supported by a VIA reviewer. Smartphone-based training and mentorship were perceived by providers as important and essential complementary processes to further develop diagnostic and management competencies [25]. In semi-rural Tanzania, five providers were trained to perform smartphone-enhanced VIA with real-time trainees supported by regional experts. Images were sent through smartphone applications on the available mobile telephone networks. Within one month of training, the agreement rate between trainees and expert reviewers was 96.8% [26]. Maintaining competencies and accuracies of VIA are also major challenges as the standard short-term onsite VIA trainings may not guarantee skills retention. VIA web-based trainings appear to be tools that can be used for continuous education, to maintain frontline healthcare providers’ skills, which will eventually contribute to a higher diagnosis performance [27,28] (Table 2).

## 7. Off-Site Mentorship

The use of D-VIA enables screeners from remote areas to send cervix images to a more experienced supervisor, ensuring a quality assurance system. Recent studies showed that off-site detection of cervical lesions based on evaluation of smartphone photographs is feasible in a low-resource context and seems to be as reliable as an on-site diagnosis [22,23]. D-VIA based on cellphone screening allows communication between frontline healthcare providers working in remote clinics and experts located in urban area, thereby allowing support and concertation between physicians. In Tanzania, a project that is currently underway, “The Kilimanjaro Cervical Screening Project”, in which healthcare workers perform screenings by VIA and send images through their mobile phones for confidential, prompt, expert consultations, open up new dimensions to the VIA approach. Coupling smartphones to VIA can be of significant contributions in improving quality work and reducing patients’ waiting time and loss of follow-up [26]. (Table 1 and Table 2).

In Cameroon, the Extension for Community Health Outcome (ECHO) telementoring scheme was implemented to provide support to nurses running the Women’s Health Program of the Baptist Health Services [35]. It is a program run by trained nurses to provide lower-cost cervical cancer screenings and treatments. During meetings using the ECHO model, specialists from the USA, Canada, Europe, and Africa have reviewed clinical cases presented by front line nurses and provided tailored didactic lectures. This approach is highly appreciated by frontline providers for the opportunity to learn with peers. They reported that ECHO sessions increased their ability to access specialty care.

## 8. Data Registration and Monitoring

Important advantages of D-VIA include the following: it allows storing patients’ cervical images and data on online databased, which facilitates communication between health professionals, during patient follow-ups, and prevents the loss of information. In Madagascar, a digitalized patient record, the Cervical Cancer Prevention System (CCPS), was developed. It is a mobile health application that allows registration of clinical data, which can be transmitted onto a secure, web-based platform using an internet connection [31]. Healthcare providers have access to the central database and can use it for follow-up visits. In rural India, Bhatt et al. showed the use of a mobile health system as a key component for cervical and oral cancer screenings [36]. Data were entered manually and the application contained permission checks to minimize the risks of “personal-identifying” data security breaches. Peterson et al. showed that the EVA system, consisting of “a mobile colposcope built around a smartphone” with “an online image portal for storing and annotating images”, could monitor real-time screening data and identify positive patients to organize treatments when required [34] (Table 2).

## 9. Artificial Intelligence: A Vision of the Future

Artificial intelligence (AI) is revolutionizing medical diagnoses, and applications in the field of cervical cancer screening have demonstrated that it can achieve higher diagnostic accuracies than experts [37,38]. Most AI tools for cervical cancer screenings have been developed using images taken by standard colposcopes [39,40,41,42]. Nevertheless, colposcopes are rarely available in low-resource settings and, therefore, AI models are being developed using images captured by other acquisition devices more adequate for limited resource contexts. In particular, smartphones are used as acquisition devices for the development of algorithms, with advantages described above.

Mobile ODT uses smartphones with optical lenses attached to acquire images and their databases are used to develop classification algorithms, such as in [43], proving the potential of using smartphones as acquisition devices for developing automatic diagnosis tools. Kudva et al. used an android device with a camera to develop AI classification tools, achieving very high performances [44]. Bae et al. developed a smartphone-based endoscope that acquires and classifies images before and after application of the acetic acid [45]. Finally, Viñals et al. developed an AI algorithm based on smartphone images without any additional component [46]. In the near future, smartphone mHealth and automated visual evaluations may allow for the automated and accurate detections of CIN and become key elements in cervical cancer screenings. (Table 3)

## 10. Conclusions and Perspective

Concerns about data protection are sometimes addressed when it comes to smartphone-use in a medical context. Smartphones and any specific app used in cervical cancer screening programs are considered medical devices. The WHO considers medical devices as “indispensable to advance universal health coverage, address health emergencies, and promote healthier populations”; and most countries have regulatory controls and regional guidelines for the correct use of medical devices and mHealth data. As in every study protocol approved by an ethics committee, patients’ agreement is necessary for the use of personal data.

Smartphone D-VIA is a promising tool used to improve the quality and efficiency of cervical visual assessment in low resource settings and in remote areas; therefore, helping clinicians in the diagnosis of pre-cancerous lesions [17]. Images can be stored in a VIA image bank and be used for training; sharing real-time images with long-distance experts will improve the quality of work of healthcare providers. Although the evidence supports that D-VIA improves CIN2+ diagnostic performance, the use of smartphone applications is only considered as a tool to minimize the subjectivity of the diagnosis. The use of mHealth applications is on the rise and might improve and facilitate cervical cancer screening by guiding healthcare workers through a decision-making algorithm, independent of the level of experience. In the future, a computer-assisted automated visual evaluation will be able to discriminate between normal and CIN and will likely significantly improve diagnostic accuracies, as well as allow see-and-treat approaches.

LMIC healthcare providers should focus on the implementation and development of smartphone-based screening programs using D-VIA, as it is proven to be acceptable and inexpensive, and it aligns with the WHO’s effort towards elimination of cervical cancer in the twenty-first century. Rossman et al. published a systematic review on the use of digital health strategies for CC control in LMICs, showing that most interventions focused on secondary prevention [47]. Strategies are used to “facilitate patient education, digital cervicography, health worker training, and data quality”, but the evidence for effectiveness is limited and comes under a lot of bias. A meta-analysis could not be conducted because of the lack of matched outcomes between studies, which support the need to conduct further and stronger studies in developing countries.

## Figures and Tables

**Figure 1 healthcare-10-00391-f001:**
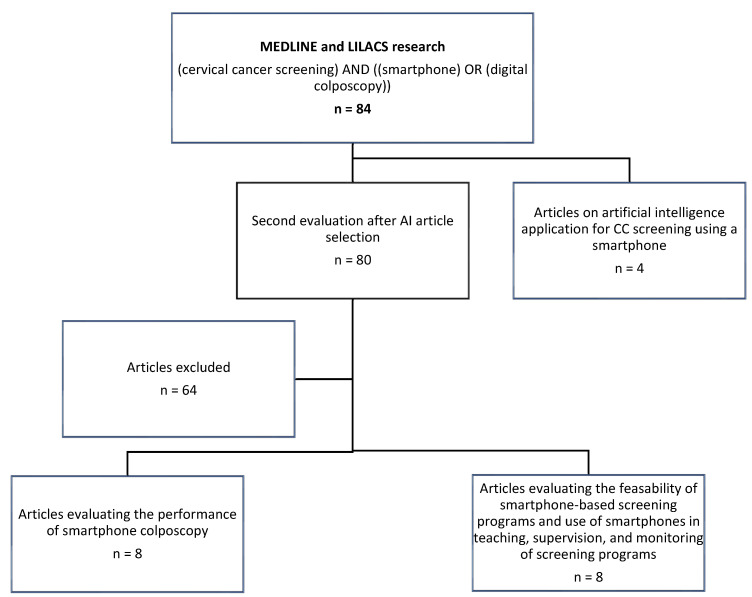
Flowchart of articles selection.

**Table 1 healthcare-10-00391-t001:** Studies evaluating the performance of a digital colposcopy using a smartphone for cervical cancer screening in LMIC.

Study	Population	Intervention and Device	Outcome and Results	Strengths and Weaknesses
Mungo et al., 2021 [19]	Western Kenya 25–49 y HIV-positive *n* = 164 *	D-VIA images taken by nonphysicians. Samsung J8; three off-site expert colposcopists assessed images.	Outcome: performance to detect CIN2+ (off-site) and acceptability of D-VIA. Results: Se ranging from 21.4% (95% CI, 0.06 to 0.43) to 35.7% (95% CI, 0.26 to 0.46). Sp between 85.5% (95% CI, 0.81 to 0.90) to 94.9% (95% CI, 0.92 to 0.98). 99.4% of women were comfortable with the use of a smartphone.	Comment: low sensitivity, very good acceptability. Strengths: histology as reference standard. Limitations: HIV population.
Goldstein et al., 2019 [20]	China (rural Yunnan areas) 35–65 y *n* = 216 *	VIA and digital images. Samsung Galaxy J5 Pro (mobile ODT system).	Outcome: performance to detect CIN1 and CIN2+ Results: Se: NR, Sp: NR.	Comment: accuracy of D-VIA to differentiate between CIN1 and CIN2+ Strengths: histology as reference standard. Limitations: low observed prevalence of HPV (6%), small number of CIN2+ (*n* = 15).
Thay et al., 2019 [21]	Cambodia 30–49 y *n*= 250 HPV-positive = 56 **	VIA and digital images. Samsung Galaxy J5 Pro (mobile ODT system).	Outcome: differentiation between CIN1 and CIN2+. Results: Se: NR, Sp: NR.	Comment: accuracy of D-VIA to differentiate between CIN1 and CIN2+. Strengths: histology as reference standard (but only in case of CIN2+ suspicion). Limitations: study setting in an urban hospital, results might not be applicable to rural regions, few CIN2+ lesion (*n* = 4).
Tran et al., 2018 [18]	Madagascar 30–69 y *n* = 125 *	Forty-five gynecologists (different levels of expertise) assessed D-VIA images. Smartphone Galaxy S4/S5.	Outcome: performance to detect CIN2+. Results: Se 71.3% (95% CI 67–75.7); Sp 62.4% (95% CI 57.5–67.4)	Comment: visual assessment demonstrated relatively high Se. Strengths: histology as reference standard. Limitations: small sample size (19 CIN2+).
Gallay et al., 2017 [17]	Madagascar 30–65 y *n*= 56 *	Four clinicians assessed D-VIA images and classified them in an app called “Exam”. Smartphone Galaxy S4/S5.	Outcome: evaluation of image quality and inter-observer agreement. Results: adequate quality for visual assessment in 93.3% of cases. Moderate inter-observer agreement, with kappa value = 0.45 (0.23–0.56).	Comment: small study, designed only for quality of images. Limitations: no histology for diagnosis confirmation.
Urner et al., 2017 [22]	Madagascar 30–69 y *n* = 187 *	Fifteen clinician evaluated D-VIA images (off-site). Samsung Galaxy S4/S5.	Outcome: performance in the detection of CIN2+. Results: Se 94.1% (95%CI 81.6–98.3); Sp 50.4% (95%CI 35.9–64.8).	Comment: Se to detect CIN2+ lesion better than generally reported. Strengths: histology as reference. Limitations: small sample size and limited number of CIN2+ (*n* = 14).
Catarino et al., 2015 [23]	Madagascar 30–65 y *n* = 137 *	Comparison of VIA (on-site) and D-VIA (off-site). Samsung Galaxy S4/S5.	Outcome: performance to detect CIN2+ and inter-observer agreement. Results on-sites: Se 66.7% (95%CI 30–90.3); Sp 85.7% (95%CI 76.7–91.6). Results off-site: Se 66.7% (95%CI 30–90.3); Sp 82.3% (95%CI72.4–89.1). Moderate to poor inter-observer agreement: kappa 0.28.	Comment: higher Sp than generally reported, demonstration that off-site assessment is feasible. Strengths: histology as reference Limitations: 30.7% drop-out rate, small sample size
Ricard-Gauthier et al., 2015 [16]	Madagascar 30–65 y *n* = 122 *	Comparison of VIA and D-VIA (on-site) and D-VIA (off-site). Samsung Galaxy S4.	Outcome: performance to detect CIN 2+. Results on-site: Se 28.6% (95%CI 3.7–71%), Sp 87.2% (95%CI 77.7–93.7%). Results off-site: Se ranging from 42.9 (95%CI 9.9–81.6) to 85.7% (95%CI 42.1–99.6); Sp from 48.1 (95% CI 38.5–59.7) to 79.2% (95%CI 68.5–87.6).	Comment; Off-site assessment feasible, lower Se for on-site assessment than reported in literature. Strengths: histology as reference. Limitations: 27.9% drop-out rate, small sample size

Abbreviations: CIN (cervical intraepithelial neoplasia), DC (digital colposcopy), D-VIA (smartphone-based visual inspection with acetic acid), D-VILI (smartphone-based visual inspection with Lugol iodine), ECC (Endocervical curettage), HPV–Hr (human papilloma virus–high risk), HPV-positive (human papilloma virus positive), NR (not reported), Se (sensitivity), Sp (specificity), y (years old). * All HPV-positive; ** 56/250 women were HPV-positive.

**Table 2 healthcare-10-00391-t002:** Studies evaluating feasibility of smartphone-based screening program in LMICs, staff training, and on/off-site supervision.

Studies	Population	Intervention	Results, Comment
Asgary et al., 2020 [29]	Eswatini 25–49 y *n* = 247 HPV status = NR HIV-positive = 128 *	Smartphone-based VIA screening program, standard VIA training, refresher course, and 6-month mHealth mentorship.	Results: agreement 100% for positive cases and 95.7% for negative; kappa 0.74, then 0.64 after 3 months and 0.79 after 6 months,
Yeates et al., 2020 [30]	Tanzania >24 y *n* = 10,545 HPV status = NR HIV-positive = 2561 **	Smartphone-enhanced VIA platform (SEVIA) for “real-time secure sharing of cervical images”. Follow-up of the mean VIA+ rates after implementation of SEVIA. Evaluation of VIA images by providers and reviewers.	Results: VIA+ rates increased from 4 to 6.2% after implementation of SEVIA. Provider-Reviewer concordance rate = 90% over the 1-year period. Comment: SEVIA allows enhanced quality of visual inspection, training, real-time data acquisition, monitoring, and evaluation.
Asgary et al., 2019 [25]	Ghana mean age = 33.8 y *n* = 21 HPV status = NR	Providers’ perceptions and experiences: 15 nurses, 1 nurse supervisor, 1 expert reviewer.	Comment: cervical images provided peer-to-peer learning opportunities, better trust of patients, targeted education, and improvement of adherence, as well as implementation of quality control.
Quercia et al., 2018 [31]	Madagascar 30–65 y *n*= 151 HPV status = NR	Registration of cervical cancer screening program data onto a secure web-based platform, for monitoring purposes. Quality of data evaluation.	Results: less than 0.02% of key data missing. Comment: small group. Helps for real-time monitoring, but impact on women follow-up not assessed.
Sharma et al., 2018 [32]	India Mean age = 38.79 y *n* = 180 HPV status = NR	Assessment of nurses’ judgment for diagnosis of cervical pre-cancerous lesions using smartphone images.	Results: moderate nurse-expert agreement, kappa 0.45. Comment: appropriately trained nurses can reliably conduct screening. Real-time expert feedback might improve reporting.
Asgary et al., 2016 [33]	Ghana 25–45 y *n* = 169 HPV status = NR	Providers completed a 2-week on-site training in VIA, followed by a 3-month VIA training supported by text messaging by an expert reviewer (real-time feedback). Comparison of agreement rates for VIA+.	Results: total agreement rate, 95%, average agreement rate between each provider and expert reviewer 89.6%. Kappa 0.67
Peterson et al., 2016 [34]	Kenya Age = NR *n* = 824 HPV status = NR	Training of providers using pictures taken. Decision support “Job Aid tool” included in the mobile application (MobileODT system) for diagnosis and treatment.	Results: 12.6% pre-cancerous lesions, 0.7% suspected cancer. Comment: deployment of the “EVA System” allows monitoring of clinical decisions made by nurses. Help of “Job Aid” decision support for treatment and gives more confidence to providers.
Yeates et al., 2016 [26]	Tanzania 25–49 y *n* = 1072 HPV status = NR	Training providers to perform D-VIA with real-time support from regional experts, images sent through a smartphone application.	Feasibility of smartphone camera to perform “Enhanced VIA” and level of agreement between trainee and expert over time (agreement 96.8%), Response timing (real-time), 1–5 min 48.4% and <10 min 60% of the time.

Abbreviations: D-VIA (smartphone-based visual inspection with acetic acid), EVA (enhanced visual assessment), HIV (human immunodeficiency virus), HPV (human papilloma virus), mHealth (mobile health), NR (not reported), SEVIA (smartphone enhanced visual assessment), VIA (visual inspection with acetic acid), y (years old). * 128/247 women were HIV-positive. ** 2561/10,545 women were HIV-positive.

**Table 3 healthcare-10-00391-t003:** Articles on artificial intelligence application for CC screening.

Studies	Population	Objective	Device	Intervention	Results
Kudva et al., 2018 [44]	India >24 y *n* = 102	Develop a decision support system for cervical cancer screening with an inbuilt image processing algorithm.	Android device with a camera of 13 Mpx.	102 images Reference = expert evaluation.	Accuracy 97.9%, Se 99.0%, Sp 97.1%, AUC NR.
Bae et al., 2020 [45]	South Korea, >20 y *n* = 20	Develop a new cervical cancer screening technique and implement a machine-learning algorithm using images taken during VIA with a smartphone-based endoscope.	Smartphone-based endoscope.	40 images (2 per patient). Expert evaluation vs AI. Reference = histopathology.	Accuracy 78.3%, Se 75.8%, Sp 80.3%, AUC 0.805. Clinicians’ mean accuracy 77.5%, Se 62.5%, Sp 100%, AUC NR.
Xue Z. et al., 2020 [43]	Various countries >18 y *n* = 3221	Evaluate accuracy of automated visual evaluation (AVE) on smartphone images.	MobileODT system (smartphone with lens).	7587 images. Reference = expert evaluation	Accuracy NR, Se NR, Sp NR, AUC 0.87 (95% CI 0.81–0.92).
Viñals et al., 2021 [46]	Cameroon, Switzerland 30–49 y *n* = 44	Development of a smartphone-based algorithm to detect cervical precancer from the dynamic features (dynamics of aceto-whitening).	Samsung Galaxy S5	44 dynamic images; Expert evaluation vs. AI. Reference = histology	AI accuracy 89%, Se 90%, Sp 87%, AUC NR. Clinicians’ mean accuracy 71%, Se 68%, Sp 78%, AUC NR.

Abbreviations: AI (artificial intelligence), AUC (area under the curve), Mpx (megapixels), NR (not reported), Se (sensitivity), Sp (specificity), VIA (visual inspection with acetic acid).

## Data Availability

Not applicable.

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
