# Peer review of "Smartphone-Based Visual Inspection with Acetic Acid: An Innovative Tool to Improve Cervical Cancer Screening in Low-Resource Setting"

_healthcare, 2022, doi:10.3390/healthcare10020391_

Round 1
Reviewer 1 Report
In the review titled “Smartphone-based Visual Inspection with Acetic Acid: an Innovative Tool to Improve Cervical Cancer Screening in LowResource Setting”, the authors provide a survey of studies that have used smartphone devices to conduct digital visual inspection with acetic acid for cervical cancer screening in low resource settings. They have cited factors from these studies that emphasize the superiority of digitally enhanced VIA over traditional VIA inspection, in terms of specificity and sensitivity of detection. It is clear that not only does D-VIA provide accuracy of screening, but also other benefits like off-site screening, data monitoring and ease of use without the need for expensive equipment.
The review is well written, provides sufficient background and lists the strengths and weaknesses of the approach in a systematic manner.
Minor concern: It seems that Table 3 is mislabeled as Table 1 in the manuscript. Please rectify.
Author Response
Thank you very much for your interest in this article and for your positive comments.
Minor concern: It seems that Table 3 is mislabeled as Table 1 in the manuscript. Please rectify.
Comment: We have made the rectification.
Reviewer 2 Report
This review paper is very interesting and could be used by physicians to have a general overview of the technique.
I would recommend to include a short section devoted to the evaluation of the efficiency of VIA against standard methods used in medium and high income countries.
I also would strongly recommend to include a section discussing the regulatory issues involving the use of smart phones and apps in medicine. Smart-phones, connectivity and software are undoubtedly very powerful tools. However, at the moment a smart-phone or an app is used in the medical domain they are (by law) considered as medical devices. Both, the smart-phone and the app are now medical devices. Therefore, they are under the jurisdiction of the regulatory agency corresponding to the country were the smart-phone and/or the app is applied to patient.
For example: The FDA has jurisdiction over medical devices in the USA and must approve all medical devices before they are made available to the public. The same apply to any other country.
Please consult:
1) Device Software Functions Including Mobile Medical Applications. in https://www.fda.gov/medical-devices/digital-health-center-excellence/device-software-functions-including-mobile-medical-applications
2) Authors may also consult mHealth (mobile health) FDA regulatory. Including the use of cell phones, smart-phones, etc.
Author Response
I would recommend to include a short section devoted to the evaluation of the efficiency of VIA against standard methods used in medium and high income countries.
Comment: Indeed, it is interesting to share this information in our review; we added a paragraph in the VIA: Strenghts and limitations section about the efficiency of VIA against HPV testing and cytology in high income countries.
I also would strongly recommend to include a section discussing the regulatory issues involving the use of smart phones and apps in medicine. Smart-phones, connectivity and software are undoubtedly very powerful tools. However, at the moment a smart-phone or an app is used in the medical domain they are (by law) considered as medical devices. Both, the smart-phone and the app are now medical devices. Therefore, they are under the jurisdiction of the regulatory agency corresponding to the country were the smart-phone and/or the app is applied to patient.
Comment: Thank you for this relevant comment, we have added a paragraph in the conclusion and perspective section to address this issue.
Reviewer 3 Report
This study was an interesting review that provides useful data.
I suggest that the author adjust and included in the tables similar results for better contrast.
For example:
Table 1, some reports do not include Se and Sp and others have more information, such as CI (and the authors do not dispute these data). I suggest that authors included a discussion about the next questions:
What is the mean of sensitivity and specificity in all the reviewed works?
And if exist discrepancies between works, what is the reason?
What percentage of studies reviews, focused to detected CNI1 and CNI2, obtained positive results?
Table 2.
I suggest that the author adjust and included in the tables similar results “rates increased or concordance rate” to better contrast.
Please include a discussion about the next questions:
What percentage increased the training to obtain a higher diagnosis, according to all review works?
What is the minimum training time that reflects improvement in the diagnosis? (Consider the average of all the works performed
Minor
Table 1
In the work of Mungo et al, 2021 is absent the asterisk, but in the text, the authors indicated that “all HIV and HPV-positive”
In the work of Goldstein et al, 2019 the population number is confused because the author indicated “limitations: low observed prevalence of HPV (6%), however, the author also indicated HPV positive n=216* ( *all HPV-positive)
Author Response
- Table 1
Some reports do not include Se and Sp and others have more information, such as CI (and the authors do not dispute these data). I suggest that authors included a discussion about the next questions:
What is the mean of sensitivity and specificity in all the reviewed works?
And if exist discrepancies between works, what is the reason?
What percentage of studies reviews, focused to detected CNI1 and CNI2, obtained positive results?
Comment: This review is not intended to be a systematic review but a scoping review; our aim was to include all studies meeting our criterias to have an overview of what Smartphone can bring to cervical cancer screening in LMICs, and what can be implemented or developed. Therefore, studies we included have all different methodology and outcomes. Calculating a mean sensitivity and specificity do not seem to be relevant in those conditions, especially with on- and off-site evaluation.
We added a sentence to explain this point.
- Table 2.
I suggest that the author adjust and included in the tables similar results “rates increased or concordance rate” to better contrast.
Comment: studies that evaluated agreement rate between nurse-expert or on- and off-site expert for CIN1/2 diagnosis did not use any comparison; contrast is therefore hard to show.
Please include a discussion about the next questions:
What percentage increased the training to obtain a higher diagnosis, according to all review works?
What is the minimum training time that reflects improvement in the diagnosis? (Consider the average of all the works performed
Comment: there is only one study (Asgary et al, 2020 [43]), that calculated the difference in agreement rate over time. In all the works performed, the training time for non-experts was between 1 and 6 months.
Minor
In the work of Mungo et al, 2021 is absent the asterisk, but in the text, the authors indicated that “all HIV and HPV-positive”
Comment: For more clarity, only the asterisk was left to note HPV-positive women
In the work of Goldstein et al, 2019 the population number is confused because the author indicated “limitations: low observed prevalence of HPV (6%), however, the author also indicated HPV positive n=216* ( *all HPV-positive)
Comment: Total population number N = 3600, of which 216 were HPV positive (6%) and those were included in the protocol
Round 2
Reviewer 2 Report
Dear Authors;
Thank you very much for improving the manuscript. I consider the article is ready for publication.
sincerely
This manuscript is a resubmission of an earlier submission. The following is a list of the peer review reports and author responses from that submission.
Round 1
Reviewer 1 Report
Your introduction appropriately cites the WHO goal of having 70% of women screened with a high performance test. I would follow that with a note that VIA is not a high performance test.
In each of the tables, you list the performance characteristics (sensitivity, specificity, etc.) of the technologies discussed. Discussion of these should be included in the text as well.
Section 3 is about strengths and limitations of VIA. I expected to see some comment on sensitivities and specificities. There is a lot of literature, some favorable, some not so. You state that VIA is considered by some to be a low standard of care, yet some of the literature disagrees. The paper would be stronger with discussion of some of this literature with sensitivities and specificities of VIA from a few studies included.
Section 3.3 In the third paragraph, discussing training in Tanzania, you mention that within 1 month agreement between trainees and trainers is 96.8%. Does the study offer any control or comparison parameter, such as agreement before training or in a control group?
Section 3.4 In discussing the Kilimanjaro Project, you refer to sending images over mobile phones as a "new dimension in the VIA approach." This is an adaptation for VIA of the technique mobile ODT has been marketing for colposcopy for years. Consider modifying the phrase "new dimension."
Section 4. Most of the section discusses AI technology without any documentation of accuracy. (Again, it's in the table but should be in the text.) The last sentence makes the statement that "mHealth and automated visual evaluation will allow the automated and accurate detection..." The text says nothing to substantiate that prediction other than that it's in development. Consider changing "will allow" to "may allow."
Section 5. Conclusions Sentence 3 states that the available data supports the conclusion that D-VIA may improve diagnostic performance. Please make clear if you're comparing D-VIA with VIA alone. Also somewhere in the paper, some comment should be made as to whether the studies referenced were for screening VIA or VIA triage after positive HPV. If they referred to screening VIA, then some comment should be made as to whether the results could be generalized to VIA following positive HPV or if more study is needed.
Reviewer 2 Report
This paper provided a summary of some of the data on the use of smartphone based VIA in LMIC. Overall there are summaries of many of the approaches being taken in LMIC which will be of interest to readers. The major concern of this paper is the disconnect between the methodology and the results. In some ways this is framed as a systematic review however this is not borne out in the methodology and several of the results presented are not reflected of what is described in the methodology presented. I would suggest this paper needs to be reframed as a commentary rather than as a review, or as a limited review rather than as a systematic review with a search strategy etc as presented in the methodology.
Title / Abstract:
- It is not clear from the title or abstract whether this was conducted as a systematic review. If it was then the PRISMA statement should be supplied and followed, which would include identification that the manuscript is a systematic review in both the title and abstract.
- In addition other details in the abstract that should be included are the inclusion / exclusion criteria etc. (https://www.bmj.com/content/372/bmj.n71).
- If this is not intended as a systematic review and rather say a scoping review, then the abstract should justify in some way why only medline was searched, which may seriously limit the capacity to meet the objective of the use of D-VIA in LMIC given it excludes databases unique to LMIC such as LILACS.
Introduction
- Reference 1 could be updated to the 2020 GLOBOCAN data
- Reference 2 and 3 need formatting in the reference list
- Common parlance is to say "WHO" rather than "the WHO"
- The abstract refers to both smartphone VIA as well as the use of artificial intelligence. There is no background provided on AI in the introduction - this should be added
- Again the aim of the manuscript is stated here to review the data available on smartphone use - there are significant concerns with the capacity to meet this objective if only one database was search
Methodology
- A serious limitation here is the use of only one database. There are potentially several papers that may have met the inclusion criteria that are missed (e.g. Rashmi B, Vanita S, Radhika S, Niranjan K, Payal K, Sarif K, et al. Feasibility of using mobile smartphone camera as an imaging device for screening of cervical cancer in a low-resource setting. J Postgrad Med Edu Res. 2016;50:69-74 or Sharma D, Rohilla L, Bagga R, Srinivasan R, Jindal HA, Sharma N, et al. Feasibility of implementing cervical cancer screening program using smartphone imaging as a training aid for nurses in rural India. Public Health Nurs. 2018;35(6):526-33.). The authors should both justify the use of only one database as well as justify the use of limited search teams (was a sensitivity analysis with these terms conducted?)
- The inclusion criteria states that studies were excluded if they were not conducted in a low income country however the title of table 1 states the studies are those from LMIC and certainly middle income countries such as China are included. The search strategy and inclusion criteria need to be clarified.
- The authors interchangeably use low resource and low income - perhaps one or the other for consistency would be ideal
- The medline search given seems somewhat limited. There are significantly more mesh terms for cervical cancer screening and for smartphone which are likely to result in a larger number of articles for screening and may increase the number of articles included. A librarian may be of assistance here? Again I would strongly encourage the use of more than one database
- The reasons for exclusion should be justified
Section on VIA
- Much of section 3 and the beginning of 3.1 is about VIA and digital photography rather than smartphone use, which is the objective stated. I think most of this information could be compressed and moved to the introduction / background rather than presented in the results section. Alternately if the aim is to present the data on digital cervicography than the methods should be updated to reflect this
- MobileODT should be referenced
- Table 1 refers to the data on performance for the detection of CIN 2+. The detection of CIN 2+ should be both defined and used as an inclusion criteria in the methodology
- Again a paper on digital cervicography is introduced (Firnharber et al) - this belongs in the introduction rather than the results section, or a change in methodology needs to occur.
- Overall the text under the title "performance of CIN2+ diagnosis (table 1)" does not reflect what is in table 1. For example the authors discuss a paper by Gallay et al which is not included in table 1 - is this background? should it have been included?
Section 3.3 (table 2)
- It is not clear in this section or from the data in table 2 whether these were all papers that would technically meet the inclusion criteria describe in the methodology? The inclusion criteria included that papers had to be conducted in a cervical screening context however some of the papers here appear to be in a training context only?
Section 3.4
- Data is referred to in this section which is both not in table 2 nor included in the methodology. For example the authors discuss the Cameroon ECHO project. Overall there remains a disconnect between the methodology and the results presented
Section 4
- AI is introduced although not considered in the inclusion criteria or the search strategy
- Data included in table 3 includes some that is obtained in Switzerland - which is not an LMIC
Reviewer 3 Report
Thank you for the opportunity to review this interesting paper entitled “. Smartphone-based Visual Inspection with Acetic Acid: An Innovative Tool to Improve Cervical Cancer Screening in Low-Resource Settings.”
This paper presents interesting data and is informative. I command the authors for this work.
However, this paper needs some important clarification on the content, methods, results and discussion.
General comments: Some of the statements appear to be directly taken from some studies that were cited. I recommend addressing these and or provide direct citations and put inside of quotation marks.
The format of presentation is a bit unusual as it does not provide any Results or Discussion sections. Instead, it is sub-headed by numbers into 1.Introduction, 2.Methodology, different sections of screening tests, and then Conclusion and perspective. This format might be from the journal’s instruction but if not, it needs to conform to the standard presentation for better comprehension and flow.
Specific comments:
Abstract:
Change “poorly documented” to “ is not well-documented”
Suggest changing “low-income countries” to “low-resource settings” and keep throughput the manuscript for consistency.
Why was January 2015 selected as the start date? Provide justification. Some of your citations and studies you have cited are from before 2015.
The discussion of artificial intelligence is a bit out of place. Conform it to the results and conclusion.
Introduction:
Age 35-45 is different in the current WHO recommendations. You may want to update.
Para 2: Study by Asgary et al in Eswatini also documented the subjectivity and loss of competency over time.
Para 3: Last sentence needs citation.
Para 4: The entire paragraph until “Our aim” needs citations. First, second, third and fourth sentences of para 4 all should have citations. The first few sentences don’t seem to belong here but to the conclusion or discussion of this paper, and should not be in the introduction.
Methods:
Have you the authors different MeSH terms of used terms in these categories? For example, what if a study used “mobile phone” as opposed to Smartphone? Would your search have captured that?
There is no any term regarding Artificial Intelligence. How did the authors do that search then?
Why did not the authors use grey literature database search along with Medline? Many reports or evaluations of services by NGOs or Foundations providing services on cervical cancer screening are likely not published in the Medline.
Why did not the authors use VIA as a search term? It is unusual to have a review on VIA and not actually using this term in searching databases. Is not VIA the basis for this review?
The search did not have “low-resource setting” and authors did not use VIA which is the main screening methods in low resource settings.
What did the reviewers review? Title and abstract? Or the entire texts of 65 papers? Clarify.
Figue1: What were major reasons for the exclusion of 47?
Figure 1: The authors should not add AI to this diagram because it was conducted separately as they described and no search terms for AI were used.
How did the authors define what is feasibility and use of smartphone in teaching? Why is there no section of “smartphone-VIA” as opposed to “smartphone colposcopy”? WHO has not recommended smartphone colposcopy or colposcopy as screening methods? It has recommended VIA.
Section 3:
This sentence comes out of nowhere because it is not included in the figure.
The entire section does not seem to be part of the Results. Instead, it should go in the introduction to describe the VIA and its process as a screening method. In that case, it also needs citation.
Citation 9 in the second para in this section is irrelevant to this sentence.
This entire section 3 seems to belong to introduction.
Section 3.1: Study from 2019 is not relevant to this paper because it is not about Smartphone VIA which is the title of this review.
There are at least 4-5 other studies that are specific to the use of digital stationary camera for VIA.
Citation 13 and 14 are also less relevant to this section.
Last sentence of section 3.1, para 1, needs citation.
Second para of section 3.1: Asgary et al (citation 53) have already documented this in 2016 without the use of MobileODT in 2016. Some of terminologies and phrases used here appear to be directly from this and other study and should be cited.
Section 3.2:
First para: Please describe how the accuracy of D-VIA was measured by the article’s authors? Against Pap testing? Against Colposcopy?
Second para, the study from Madagascar, talks about feasibility and usability and not performance of VIA for CIN 2+ diagnosis. This study is only for quality of images and not accuracy of diagnosis the way it is presented. It might not belong here to this section.
Third para: First sentence needs citation. Are authors talking about citations in Table 1?
Could the authors please summarize the finding of Table 1 under this section of 3.2?
It appears that studies by Yeates and Asgary et al should be under this section because they used the expert reviewer as the gold standard for assessment of smartphone-based VIA. Asgary et al in Ghana and Eswatini got Sens/Spec, NPV, PPV. In ESwatini they showed the impact on reliability and reproducibility of VIA programs and as a quality process.
Section 3.3:
Table 3: Asgary et al (50), results of this study included peer-to-peer education and peer support via weekly team meetings and reviewing of imaging and medical records of the images. Intervention included remote mentorship via telehealth and image review by the expert and quality control. Intervention also included development of smartphone photography guide. The program started with usual VIA training, a follow up period with assessment of performance, then refresher training and follow up period, and then smartphone-based VIA training and mentorship. VIA results and performance were compared between phases. In Eswatini it showed the impact on reliability and reproducibility of VIA programs and as a quality process. Authors could consider updating the table with this information.
First para of section 3.3, last sentence, should cite Asgary et al from 2016 and 2020 and Yeates from 2020.
Second para of section 3.3, needs to add findings from patients who underwent smartphone imaging including lack of stigma and no privacy concerns, good acceptability by patients and that they took their pictures for partners to educate them. It is also important to state that the study did not face important logistical barriers for smartphone and network connectivity and that related barriers were minimal or overcame without significant resources. That expert review was feasible and appreciated by providers and patients.
Section 3.4:
Para 1: Both studies by Asgary et al (53 and 50) implemented and evaluated the impact of off-site mentorship and supervision and should be described and cited here. Images were taken from VIA (before and after) and sent through a closed loop secure telemedicine to the expert reviewer in Eswatinin study and via text messaging to the expert reviewer in Ghana study.
Asgary et al, also currently conducting a large-scale implementation study in Malawi (40,000 enrolled) using off-site mentorship and telehealth training using smartphone-based imaging (unpublished data).
Section 4.
Study by Hu et al from Costa Rica, does not seem to offer any actual verification of diagnoses. There was no use of gold standard. And there were differences in imaging methods over time and their assessment of quality and diagnostics.
Conclusion and perspective:
The first sentence is not much relevant to the paper’s topic. The idea is to compare smartphone VIA with usual VIA as an aim of this study/review. Here instead it starts by suggesting it as an alternative to colposcopy. Colposcopy is not recommended as screening methods for cervical cancer. The use of smartphone-based VIA for colposcopy is not for cervical cancer screening but perhaps for better diagnostic. If the authors recommending smartphone-VIA as an alternative to colposcopy they should clarify that it is for diagnostic scenarios/test not screening purpose.
First para should start with a discussion of findings on the accuracy and feasibility of smartphone-based VIA compared to usual standard VIA and the impact on better training for VIA, improvement of competencies, and improving of reliability and reproducibility of VIA programs. They should then cite studies from table 2.
I suggest take the discussion of smartphone colposcopy to a later section. The entire first para of the conclusion needs citations which seemed to be largely from table 2. There should also be a mentioning on the impact on quality of VIA programs through improving reliability and reproducibility of VIA diagnoses and management of cases. Meaning that the variability between providers and in one provider for different cases will decrease when smartphone- based VIA is used both for training and mentorship and as a quality control measure to assess the progress and provide feedback to the screening providers ( Yeates 2020, Asgary et al in 2016, 2018, and 2020).
Reference:
Citations in the text do not appear to be in order that they appear in the text. Consider reordering.
WHO is spelled as Organization, W.H. It should be as World Health organization.